# mTBI Biological Biomarkers as Predictors of Postconcussion Syndrome—Review

**DOI:** 10.3390/brainsci14050513

**Published:** 2024-05-18

**Authors:** Ewelina Stępniewska, Maria Kałas, Justyna Świderska, Mariusz Siemiński

**Affiliations:** Department of Emergency Medicine, Medical University of Gdansk, 80-435 Gdansk, Poland; e.stepniewska@gumed.edu.pl (E.S.); maria.kalas@gumed.edu.pl (M.K.); justyna.swiderska@gumed.edu.pl (J.Ś.)

**Keywords:** postconcussion syndrome, mild traumatic brain injury, biomarker

## Abstract

Postconcussion syndrome (PCS) is one of the leading complications that may appear in patients after mild head trauma. Every day, thousands of people, regardless of age, gender, and race, are diagnosed in emergency departments due to head injuries. Traumatic Brain Injury (TBI) is a significant public health problem, impacting an estimated 1.5 million people in the United States and up to 69 million people worldwide each year, with 80% of these cases being mild. An analysis of the available research and a systematic review were conducted to search for a solution to predicting the occurrence of postconcussion syndrome. Particular biomarkers that can be examined upon admission to the emergency department after head injury were found as possible predictive factors of PCS development. Setting one unequivocal definition of PCS is still a challenge that causes inconsistent results. Neuron Specific Enolase (NSE), Glial Fibrillary Acidic Protein (GFAP), Ubiquitin C-terminal Hydrolase-L1 (UCH-L1), Serum Protein 100 B (s100B), and tau protein are found to be the best predictors of PCS development. The presence of all mentioned biomarkers is confirmed in severe TBI. All mentioned biomarkers are used as predictors of PCS. A combined examination of NSE, GFAP, UCH-1, S100B, and tau protein should be performed to detect mTBI and predict the development of PCS.

## 1. Introduction

Postconcussion syndrome (PCS) is one of the leading complications that may appear in patients after mild head trauma. Every day, thousands of people, regardless of age, gender, and race, are diagnosed in emergency departments due to head injuries. Traumatic Brain Injury (TBI) is a significant public health problem affecting an estimated 1.5 million people every year in the United States [1] and up to 69 million people worldwide each year, with 80% of these cases being mild [2].

Despite the significant occurrence of postconcussion syndrome, which may concern 29–90% [3] of patients after a head injury, there is no unanimous definition. 

The ICD-10 clinical criteria include a history of TBI and the occurrence of three or more of the following eight symptoms: (1) headache, (2) dizziness, (3) fatigue, (4) irritability, (5) insomnia, (6) concentration or (7) memory difficulty, and (8) intolerance of stress, emotion, or alcohol [4].

The symptoms start shortly after a head injury and can be present for weeks or months. When the symptoms continue for more than six months or one year, the condition is interpreted as persistent PCS [3].

## 2. Epidemiology

Postconcussion syndrome, as the most common complication of mild traumatic brain injury, may develop in all age groups and does not depend on physical activity. Head trauma may be a consequence of injury during some sports activity or might occur at work, especially among military personnel. Another group at high risk of head trauma is the elderly, where injury can be a result of fecklessness that is connected with senectitude and increased risk of falls [5] or, in contrast, inattention in children. 

The number of postconcussion syndrome incidents is not exactly known since most individuals are not hospitalized due to complications of mTBI. 

The prevalence of hospitalized patients with mild traumatic brain injury is about 100–300/100,000 of the population. The true population-based rate is probably above 600/100,000 [6], but a significant portion of mTBI is not treated in hospitals, which is why the exact number is unknown. According to an analysis conducted in Australia, it is estimated that there are 190,000–200,000 cases of TBI per year, and 74–90% are expected to be mTBI [7]. After detailed calculations [8] conducted by emergency departments regarding patients admitted due to head trauma, it was noted that the number of TBI patients is constantly increasing, and as a result, the incidence of concussion is increasing significantly. When comparing the prevalence of concussions, the most significant changes are seen from 2002, when there were 1071 incidents per 100,000 people per year, to 2018, when it increased to 2820 per 100,000 people per year [7]. 

There are considerably more cases of brain trauma in males than females, with approximately 78.8% of injuries occurring in men and 21.2% in women [9]. An analysis of the Rivermead Post-Concussion Symptoms Questionnaire, the Pittsburgh Sleep Quality Index, and the Multidimensional Fatigue Inventory, as well as measures of depression, anxiety, and post-traumatic stress symptomatology, shows that women are at higher risk of developing PCS and the symptoms are less significant in males [10].

Recovery from mTBI is an individual matter dependent on many factors, including the mechanisms of trauma as well as injury to other organs that may cause severe complications. Despite the physical deficits, concussion is also connected with mental instability [11]. A group with a high risk of trauma and increased risk of PTSD development is military personnel [12], whose complications seriously interrupt the permanent performance of duties. Depression, anxiety, or recurrent headaches are factors that decrease life quality and increase recovery time [13]. Extensive physical activity and sports foster an elevated risk of head trauma. An analysis showed that people who developed PCS after mTBI were at lower risk of developing persistent symptoms such as fatigue and dizziness, but returning to previous activity was a challenge and involved rehabilitation [14] in some cases. 

Another factor that influences the incidence of PCS is the mechanism of trauma and the type of vehicle [15]. Patients injured in car accidents have a 54% higher risk of developing PCS than motorcyclists [16]. The reason might be the fact that wearing a helmet on a motorcycle is obligatory, which protects the head during trauma [16]. 

A large group of individuals with mTBI are pediatric patients. Children are common victims due to their high activity and unwariness. After the first year of life, minor injuries are caused by falls, whereas major injuries are the result of car accidents (including as a pedestrian) or falling from a significant height. With age, traumas are more commonly a result of bicycle or car accidents [17].

It is estimated that 144,000 junior patients are admitted to the emergency department in the US per year due to head trauma [18]. Around 90% of those cases are mild [19] and do not require hospitalization [20].

## 3. Clinical Picture

After arrival at the emergency department, patients who have experienced head trauma are assessed according to common scales. The most popular of these are the AVPU and Glasgow Coma Scale, which provide information about consciousness disorders [21], and the Canadian CT Head Rule [22] and New Orleans Rule [9], which measure the necessity of a CT scan.

The AVPU [23] scale is a simple method to assess a patient’s consciousness and responsiveness:

A—stands for alert, 

V—stands for verbal: testing the patient’s reaction to voice stimulus,

P—stands for pain: reaction to pain,

U—stands for unresponsive: patient does not react to voice and pain stimulus.

The Glasgow Coma Scale [24] (GCS) is a scale used mainly in patients after head trauma that assesses the consciousness level in response to the defined stimuli. The range is from 3 to 15, where 15 means full consciousness, and 3 is for non-reactive patients. The points are given for eye opening, verbal response, and motor response. 

For eye opening, the maximum score is 4:-4—spontaneous eye opening,-3—eye opening to verbal command,-2—eye opening to pain,-1—no eye opening.

For verbal response, the maximum score is 5:-5—orientated,-4—confused,-3—inappropriate words,-2—incomprehensible sounds,-1—no verbal response.

For motor response, the maximum score is 6:-6—obeys command,-5—localizes pain,-4—withdraws from pain,-3—flexion response to pain,-2—extension response to pain,-1—no motor response.

The Canadian CT Head Rule [22] defines which minor head injury patients need a head CT scan and which of them are at high risk of severe brain injury after trauma. The Canadian CT Head Rule consists of five high-risk factors to determine neurological intervention and two medium-risk factors to determine the requirement of a CT scan.


**Five high-risk factors:**
-GCS score < 15 at 2 h after injury,-Suspected open or suppressed skull fracture,-Any sign of basal skull fracture, for example, hemotympanum, Battle’s sign, ‘raccoon’ eyes,-Vomiting ≥ 2 episodes,-Age ≥ 65 years.



**Two medium-risk factors:**
-Amnesia before impact ≥ 30 min,-Dangerous mechanism, for example, pedestrian struck by vehicle, occupant ejected from motor vehicle.


Due to the criteria of the New Orleans Rule [25], a CT should be performed in patients with minor head injuries with any one of the listed findings: headache, vomiting, >60 years, drug or alcohol intoxication, persistent anterograde amnesia, visible trauma above clavicle, or seizure. These criteria only apply to patients who also have a GCS score of 15. 

Postconcussion syndrome occurring as a consequence of mTBI is associated with the lack of visualization of the trauma in computer tomography scans [26] and the fact that neither surgical intervention nor prolonged observation is required in the short period following the injury.

Presentation upon arrival to the emergency department varies depending on the mechanisms of the head trauma. Blunt injuries might mean certain patients require wound management or specialist consultations from the otolaryngological or ophthalmological fields [27]. 

Head and neck pain are the first symptoms that appear after brain injury, but other symptoms usually develop in a few days to a week. Approximately 10–30% of patients develop postconcussion syndrome after mTBI [28]. 

The effects of brain injury may be distinguished as intracranial and extracranial. Intracranial effects are usually caused by injury to the frontal or frontotemporal regions, depending on the mechanism of injury. Extracranial symptoms are often a consequence of neck muscle contraction from cervical root irritation or injury. 

Clinical symptoms, based on the division into intracranial or extracranial causes, can be organized into five categories: Cognitive, which contains memory deficits, attention and concentration difficulties, speech difficulties, executive dysfunction, and fine motor difficulties.Psychological, including depression, anxiety, irritability and personal changes, fatigue, and derealization.Somatosensory and vestibulocochlear dysfunction, which involves headaches, nausea and vomiting, light and sound sensitivity, hyperalgesia, and tinnitus.Visual symptoms and oculomotor dysfunction, which involves light sensitivity, blurry vision, convergence difficulty, double vision, and Horner’s syndrome.Autonomic symptoms, including fluctuation of heart rate and blood pressure, abnormalities regarding sweating and pupils, temperature dysregulation, sexual dysfunction, sleep alterations, and poor sleep efficiency.

A detailed neurological examination is an indispensable part of the diagnosis of postconcussion syndrome. During an evaluation, it is obligatory to focus on specific elements that are a consequence of the clinical symptoms listed [29]. The checklist includes: -The vestibulocochlear system,-Autonomic dysfunction symptoms,-Brainstem and cortical assessment,-Neck dysfunction,-Cognitive function.

The above-listed areas are obligatory to examine in patients with suspicion of postconcussion syndrome. Symptoms that patients present might be connected to the mechanism of injury [30], age, or mental or toxicological state. Adolescents are at higher risk of affective and habitual symptoms after an mTBI. These effects depend on gender and time of injury [31]. 

Results of the analysis of teenagers after mTBI show that girls had higher anxiety and attention disorders after past mTBI, but new mTBIs did not have much impact. In boys, aggression was noticed after new mTBI and past mTBI increased anxiety. 

The most commonly reported symptoms are headaches, which may become chronic, dizziness; fatigue, which may disrupt recovery and reduce quality of life; irritability; concentration or sleep disorders, like insomnia [32]; memory difficulties; or intolerance of stress, emotion, or alcohol. All these inconveniences, when they become chronic, are symptoms of postconcussion syndrome, which has a significant and negative influence on the quality of life of patients. 

## 4. Diagnostic Criteria and Diagnosis

Considering the limiting criteria for PCS, there are two leading definition proposals [33] from the International Classification of Diseases, the Tenth Revision, and the Diagnostic and Statistical Manual of Mental Disorders, Fourth Edition. According to the ICD, 10 clinical criteria require a history of TBI and the presence of three or more of the following eight symptoms: (1) headache, (2) dizziness, (3) fatigue, (4) irritability, (5) insomnia, (6) concentration or (7) memory difficulty, and (8) intolerance of stress, emotion, or alcohol [34]. The DSM-IV criteria are (A) history of TBI causing “significant cerebral concussion”; (B) cognitive deficit in attention and/or memory; (C) presence of at least three of eight symptoms (e.g., fatigue, sleep disturbance, headache, dizziness, irritability, affective disturbance, personality change, apathy) that appear after injury and persist for ≥3 months; (D) symptoms that begin or worsen after injury; (E) interference with social role functioning; and (F) exclusion of dementia due to head trauma and other disorders that better account for the symptoms. Criteria C and D set a symptom threshold that requires symptom onset or worsening to be contiguous to the injury, distinguishable from preexisting symptoms, and have a minimum duration [35]. A comparison of the ICD-10 and DSM-IV criteria is presented in the Table 1 below. 

A commonly used method [36] to measure the presence and severity of PCS symptoms is the Rivermead Post-Concussion Symptoms Questionnaire. The method includes a list of questions regarding somatic, cognitive, and emotional complaints following a head trauma, which may last up to several months. 

The role of the participant is to compare the severity of each symptom, taking into consideration the preinjury period through to present complaints. Answers are rated from 0—not experienced at all to 4—severe problem [37]. 

Complaints that are listed in the Rivermead Questionnaire are comparable with ICD-10 and DSM-IV criteria. The criteria include the presence of headache, dizziness, nausea or vomiting, noise sensitivity, sleep disturbance, fatigue, irritability, feeling depressed or frustrated, poor memory or concentration, slowed thinking, restlessness, blurred vision, double vision, or light sensitivity [36]. The patient scores each symptom by its severity:-0—when the symptom is not experienced at all,-1—no longer a problem,-2—mild problem,-3—moderate problem,-4—severe problem.

The symptoms are divided into two groups. The first group (RPQ-3) is based on the first three symptoms (headache, feeling of dizziness, and nausea and vomiting), and the second group (RPQ13) consists of the next 13 disorders. RPQ-3, which can be scored from 0 to 12, is connected with early signs of PCS, and if the score is high, earlier reassessment and closer monitoring are recommended.

RPQ-13 can be scored from 0 to 52 and is associated with a later cluster of symptoms, where a higher score means a greater severity of PCS symptoms. The later cluster of symptoms impacts lifestyle, participation, and psychological functioning. 

The American Congress of Rehabilitation Medicine [38] published new guidelines for mTBI that are based on six criteria referring to patients after head injury. Detection of blood biomarkers is included in the criteria for the diagnosis of mTBI.
Mechanism of injury—plausible concussion;Clinical signs—one or more of the following: loss of consciousness immediately after the injury, alteration of mental status immediately after the injury, partial or complete amnesia, or other acute neurological signs immediately after the trauma,Acute symptoms—≥2 new or worsened from the following:
-Acute subjective alterations in mental status, for example, confusion, disorientation, daze;-Physical symptoms: headache, nausea, dizziness, vision disturbances, sensitivity to light and/or noise [39];-Cognitive symptoms: feeling slowed down, ‘mental fog’, concentration difficulties, memory problems;-Emotional symptoms, e.g., emotional lability or irritability.
4.Clinical examination and laboratory findings, —cognitive, balance, or oculomotor impairment or elevated blood biomarkers indicative of intracranial injury,5.Neuroimaging—abnormalities found on CT or MRI,6.Confounding factors not better accounted for—alcohol or drug intoxication, diseases, disabilities, or symptoms prior to the injury.

Mild TBI is suspected when the patient does not meet other criteria sufficient for the diagnosis of TBI but has ≥2 acute symptoms, ≥2 clinical and/or laboratory findings, or criterion 6 is not present. Mild qualifiers cannot be used if any of the below-listed are present: -Loss of consciousness longer than 30 min,-GCS > 13 after 30 min,-Post-traumatic amnesia for longer than 24 h.

## 5. Therapeutic Options

Complications of mTBI are alarming [40] and present increasing problems that impact a large number of patients all over the world. Usually, 90% of these symptoms are temporary and resolve within up to 2 weeks, but in some cases, they last longer. Persistent PCS is defined as the presence of symptoms for 3 months. Fifteen percent of mTBI patients are diagnosed with PCS, and a small portion of them will have extended complications that will require further diagnostics and treatment [41]. In most cases, there is a symptomatic approach in the primary phase [42], including decreased physical activity, recovery, and bed rest. Sleep has an important role in the treatment process. Circadian therapy has been proposed as a form of recovery that includes sleep regulation with melatonin therapy, morning blue light therapy, evening blue light restriction, sleep apnea treatment, and omega oil supplementation [43]. Another method that was proven to be beneficial for improving well-being, psychosocial state, and mental health is cognitive rehabilitation and neurocognitive training [44], which are still being developed as a treatment line in PCS. Another option being tested is hyperbaric therapy, the role of which has not yet been proven [45]. Taking into consideration pharmacological treatment, the most commonly used drugs for post-traumatic headaches are NSAIDs, which are used worldwide, but a drug that is under investigation in PCS is amantadine [46], which has proven to be effective. 

## 6. Biomarkers in mTBI

### 6.1. Definition of Biomarker

Due to the growing prevalence of mTBI and PCT, many investigations are underway to find the perfect solution to predicting the severity of post-traumatic symptoms. As a way to improve the management of patients after mTBI, decrease the number of hospitalizations, and prevent multiple CT scans, the measurement of blood biomarkers in the acute phase was proposed [47].

A biomarker, defined by The National Institutes of Health (NIH), is a characteristic that is objectively measured and evaluated as an indication of normal biological processes, pathogenic processes, or pharmacologic responses to a therapeutic intervention [48]. The advantages of a perfect biomarker should include high sensitivity and specificity, reproducibility, low cost, and a non-invasive process [49]. When considering the above-listed characteristics, the ideal biomarker is still being searched for. 

The mTBI biomarkers can be divided into biological, imaging, and neurophysiological markers [50]. Biological samples are used as a source of biomarkers, such as protein, micro RNA (miRNA), and lipids, originating from blood serum, plasma, cerebrospinal fluid (CSF), hair follicle, saliva, and urine. 

Neurophysiological and imaging biomarkers are widely used in the diagnosis of PCS. The currently used methods include computer tomography (CT), magnetic resonance imaging (MRI), conventional MRI, functional MRI (fMRI), diffusion-weighted imaging (DWI), magnetic resonance spectroscopy (MRS), single-photon emission-computed tomography (SPECT), positron emission tomography (PET), magnetoencephalography (MEG), electroencephalography (EEG), or functional near-infrared spectroscopy (fNIRS) [51]. 

A CT scan is a fast and cost-effective method of neuroimaging commonly used in emergency departments worldwide. From its results, the severity of TBI can be distinguished and proper treatment initiated, but it cannot predict the development of PCS. Another commonly used method of imaging is MRI, where damage to the structural integrity and disrupted functional network communication in acute and subacute phases [52] of mTBI can be identified. Those changes are noted as the main factors that led to the development of PCS. 

Biomarkers are also categorized as diagnostic, prognostic, and theragnostic to understand the state of the brain injury.

The widely described biological markers include protein S100, glial fibrillary acid protein (GFAP), myelin basic protein (MBP), inflammatory proteins (e.g., IL-6, IL-8, and IL-10), tau protein, ubiquitin carboxyl-terminal esterase L1 (UCHL1), neurofilaments (NFLs), enolase 2 (NSE), neutrophil gelatinase-associated lipocalin (NGAL), and prions/plasma-soluble cellular prion (PrPC). Two of the mentioned biomarkers, GFAP and UHCL-1, were approved by the US Food and Drug Administration (FDA) in 2018 to test patients after TBI to predict concussion symptoms.

The most common biological markers are detailed below.

### 6.2. Neuron Specific Enolase (NSE)

NSE is known to be a cell-specific isoenzyme of the glycolytic enzyme enolase, and it is a highly specific marker for neurons and peripheral neuroendocrine cells. This isoenzyme was first found in neuronal cells and later was shown to be present in cells with neuroendocrine differentiation [53]. 

Due to high neuronal potential, NSE is an indicator of brain damage that causes seizures, intracranial bleeding, ischemic stroke, coma, cardiac arrest, and traumatic brain injury. NSE, as a biomarker, has proven beneficial in differential diagnostics of small-cell lung cancer, neuroendocrine tumors, and neuroblastoma. 

NSE has high specificity to the brain and was found to be released into the serum as a result of hemolysis, limiting its accuracy as a predictor of brain injury [54]. Neuron Specific Enolase, as a marker of neuronal degeneration, is used in patients after brain injury, including mTBI. NSE’s peak appears 6–12 h after the trauma. In patients who suffered moderate or severe TBI, increased levels of NSE were connected with higher mortality or prolonged neurological disabilities [55]. The prognostic value of NSE in mild TBI requires further examination and analysis. 

### 6.3. Glial Fibrillary Acidic Protein (GFAP)

GFAP is an intermediate filament protein that is highly specific for cells of astroglial lineage [56]. It is known as an astroglial marker of injury and is found in the astroglial skeleton of both white and gray brain matter [57]. It is also expressed by the Leydig cells of the testes [58]. 

The level of GFAP is low in healthy individuals, which is why it can be used as a neurodegenerative marker in Alzheimer’s disease diagnostics. Other conditions that cause the release of GFAP are neurodegenerative and non-neurodegenerative diseases. Blood GFAP levels are correlated with the clinical severity and extent of intracranial pathology in spinal cord disorders, acute CNS trauma (due to disruption of the astrocyte cytoskeleton and their activation in response to TBI) [59], and the reaction to ischemia, malignant brain tumors, and cerebrovascular events [60]. An examination of an animal model showed that after mTBI, the level of GFAP is increased in serum and CSF, but further detailed research should be performed [59]. 

### 6.4. Ubiquitin C-Terminal Hydrolase-L1 (UCH-L1)

Ubiquitin C-terminal hydrolase L1 (UCH-L1) is an extremely abundant protein in the brain where, remarkably, it is estimated to make up 1–5% of total neuronal protein. Although it only comprises 223 amino acids, it has one of the most complicated 3D knotted structures ever discovered [61]. It is involved in the process of the ubiquitination of proteins destined for degradation via the proteasomal pathway, thus playing an important role in the removal of oxidized or misfolded proteins in both normal and pathological conditions [62]. Low levels of UCH-L1 are present in healthy individuals, but it was proven that in certain conditions, for example, neurodegenerative diseases like Alzheimer’s and Parkinson’s disease, serum UCH-L1 significantly increases. It may also be a marker of neuronal loss after aneurysmal subarachnoid [63], as well as a marker of abnormal blood–brain barrier function after severe TBI. UCH-L1 levels in both cerebrospinal fluid (CSF) and serum are elevated for several days after severe TBI [62]. UCH-L1 is a protein that is released to the blood plasma due to the reaction of neuronal damage as a result of mTBI and can be detected 4 h after the injury [64]. 

### 6.5. Serum Protein 100 B (s100B)

S100 B is a calcium-channel binding protein that is highly expressed in the glial cells and Schwann cells. It has minimal concentrations in other cells and is metabolized and excreted through the kidneys. Intracellularly, S100B regularly participates in calcium hemostasis by transferring signals from other messengers. S100B also takes part in cell differentiation and cell cycle progression. In experimental conditions, it has been noticed that it has the ability to inhibit apoptosis [65]. Extracellularly administered S100B promotes neurogenesis and neuronal plasticity, performs neuro-modulating actions, and promotes processes involved in memory and learning in both normal physiology and during traumatic conditions. When secreted, S100B is found to have paracrine/autocrine trophic effects at physiological concentrations but toxic effects at higher concentrations.

Elevated S100B levels in biological fluids (CSF, blood, urine, saliva, and amniotic fluid) are thus regarded as a biomarker of pathological conditions, including perinatal brain distress, acute brain injury, psychiatric disorders, or neurodegenerative processes [66]. 

S100B has been studied as a biomarker for many types of TBI, and its main role is in calcium homeostasis, cell survival, and differentiation [50]. S100B has been found to be elevated following mild-to-severe TBI. Studies in adults have shown that transient serum elevations of these markers after mTBI correlate with abnormal cranial CT with 90–100% sensitivity and 40–65% specificity [67]. Peak S100B levels can be found in both CSF and blood 6 h after the trauma but gradually decrease. Increased levels of S100B were found in both children and adults after TBI [68].

### 6.6. Tau Protein 

Tau protein is one of the proteins that belong to a group called microtubule-associated proteins (MAPs) [69]. Polymerization of the microtubules during axonal growth is the most beneficial function of the tau protein. Due to damage to the neurons, the level of this protein was found to increase in cerebrospinal fluid and blood in both human and animal trials [70]. Elevated levels of tau protein are already well-established in neurodegenerative conditions, for example, Alzheimer’s disease, and were also measured in migraine diagnostics [71]. In TBI patients, when used as a predictor, tau protein can be detected 6 h after injury. 

Tau protein is proteolytically modified after axonal injury, and this cleavage product is known as C-tau [72]. Cleaved-tau protein undergoes examination as a single biomarker in neuronal damage and a predictor in concussion development. 

Tauopathies are neurodegenerative diseases involving the aggregation of abnormal tau protein [73]. Tauopathies include Pick Disease (PiD), Progressive Supranuclear Palsy (PSP), and Corticobasal Degeneration (CBD). Tau protein is potentially a therapeutic target due to its neuroinflammatory actions, which influence the development of neurodegenerative diseases [74]. 

## 7. Need for Prediction of Postconcussion Syndrome

Mild head injury and its complications are some of the biggest problems in developing countries. The goal is to distinguish patients at increased risk of developing postconcussion syndrome, prevent it, and start treatment in the early phase.

Currently, guidelines exist on how to separate the patients with increased possibility of concussion symptoms. The Glasgow Coma Scale and AVPU scale are the most commonly used worldwide in patients after TBI as a first-line assessment. The Canadian CT Head Rule and New Orleans Rule are common tests for the possibility of serious head trauma and provide indications for the need for a CT scan. During admission to the hospital, the most important things are a precise neurological examination and a detailed interview. The Rivermead Postconcussion Syndrome Questioner, which is used to assess the severity of symptoms as a way to diagnose PCS development, is becoming increasingly popular. All of the factors above are a part of prognostic models [75]. The goal is the ability to identify, from the large number of mTBI patients, individuals who are at risk of developing prolonged head trauma complications. The greatest hope for delivering proper care to high-risk individuals lies in the biomarkers that can be examined immediately upon arrival at the emergency department [76]. 

Table 2 presents the prognostic value of the biomarkers discussed and compares their specificity, sensitivity, AUC (area under the ROC curve), and the cut-off level (μg/L). 

### 7.1. S100b in PCS

One of the common biomarkers is S100 protein, which is used in adult and pediatric patients. It was found that cranial injury was associated with increased levels of that protein. Individuals with significant changes in their CT scan had higher S100b levels than those without cranial injury, as well as patients who developed PCS [68]. The pediatric group analysis shows that levels of S100b were also higher in orthopedic injuries, which shows that it is not specific to head traumas [77]. The level of S100 cannot be used as a screening method to decide if a CT scan should be performed and will not predict if PCS will develop [78], but forgetfulness was noted when S100 levels were elevated [79]. 

As the S100 protein is present in all healthy individuals, it proved difficult to determine the cutoff level that can be used as a guideline to diagnose a high possibility of PCS development and PCS severity. Taking into consideration the differences in PCS diagnosis criteria, the results also differ, but S100 can be used as a marker of severe injury [1]. 

The specificity of S100 is 58%, the sensitivity is 100% [80] for mTBI, and the AUC is 0.8 [81] when the cut-off level is 1.35 μg/L. The combination of S100 and tau protein was correlated with the severity of concussion symptoms [82].

### 7.2. Enolase in PCS

Enolase, as a main protein of the brain that stands between 0.4% and 2.2% of its total soluble protein, is already well-known as a marker of brain injury in death or life-long disability cases. The role in mTBI and PCS development is under investigation [55], but the relationship between high levels of enolase and persistent headaches (6 months) is noted [79]. Another finding regarding increased enolase is a decreased neuropsychological state at 2 weeks [83] and deficits in the examination at 6 weeks [84]. Despite the fact that NSE suggests acute neuronal damage and its increased level is present in blood due to hemolysis, it is not sensitive and specific enough. The specificity of NSE is 69.4%, the sensitivity is 75% for mTBI, and the AUC is 0.72 [85] when the cut-off level is 6.45 μg/L [86].

As a consequence, there is no correlation between serum level and outcome measures [54]. Individuals with chronic complications of TBI who experienced injury in the last 12 months were found to have decreased levels of NSE. The lack of NSE most likely reflects brain atrophy due to the chronic phase of severe TBI [87].

### 7.3. GFAP in PCS

Glial Fibrillary Acidic Protein (GFAP) was tested in individuals with severe head injury whose GCS escalated between 3 and 12. According to those examinations, GFAP was found to have beneficial prognostic value in unfavorable outcomes, including death, but 6-month recovery prognostics cannot be based on GFAP levels [88]. Analysis of marker results and determination of further prognosis should consist of summarizing blood tests, detailed neurological examinations, past medical history, and the mechanism of trauma. 

Individuals detected with elevated GFAP in the first hour after the injury had consciousness disorders (evaluated by the GCS) and post-traumatic changes in radiological examination and required neurosurgical interventions [89]. Elevated levels over a short period and increases during postinjury are strong predictors of severe complications, disability, and death. Elevated levels of GFAP at admission had better prognostic value than basic neurological examination, including GCS score, pupil reaction, and age. GFAP was found to be highly specific for brain injury, and its role as a predictor, diagnostic factor, and biomarker in TBI patients should be further evaluated [90]. The specificity of GFAP is 50%, the sensitivity is 64% for mTBI, and the AUC is 0.93 [91] when the cut-off level is 1.35 μg/L [63]. Elevated GFAP was associated with loss of consciousness and memory deficiencies [82].

### 7.4. UCH-L1 in PCS

A meta-analysis performed on the Ubiquitin C-terminal hydrolase in TBI showed that it could be used as a predictor in patients after head injury. The level of this marker increases significantly in individuals diagnosed with intracranial lesions. The high sensitivity in the prediction of CT changes is a significant advantage, but it lacks specificity. A major obstacle is setting the cut-off for the plasma level of UCH-L1. Tests show that its highest level is in the first 6 h and then decreases. The greatest value is found when examining the concentration immediately after the trauma [92]. Increased levels of UCH-L1 were also found in cerebrospinal fluid that was tested in severe TBI patients [93]. 

When compared to GFAP in the first 3 h after admission to the emergency department, UCH-L1 was found to be less specific in the presence of intracranial lesions [94]. The specificity of UCH-L1 is 95%, the sensitivity is 95% for mTBI, and the AUC is 0.83 [95] when the cut-off level is 0.21 μg/L [92]. Current studies show that an examination of both GFAP and UCH-L1 could be beneficial in detecting intracranial lesions and decrease the amount of CT scans performed in mTBI patients [96]. 

### 7.5. Tau Protein in PCS 

The highest levels of tau protein occur between 12 and 24 h after the trauma, but it might be present in both the acute and chronic stages. In the first phase, it indicates acute neuronal injury, and the next period is when the neurodegeneration process develops. Another important value is the elevation of tau in relation to age and gender. Women with concussions had higher levels of tau than men [97]. The specificity of tau protein is 50%, the sensitivity is 72% for mTBI, and the AUC is 0.95 [98] when the cut-off level is 1.75 μg/ [63]. The level of tau protein was found to be increased for a longer period after injury. This provides information about the presence of neurodegenerative processes and affects neurological symptoms [99]—information that could be an indicator of postconcussion severity and targeted therapy. 

Together with S100B, tau protein was found to be specific and quite sensitive for PCS in the 3-month follow-up [1]. The presence of tau protein was also detected in severe intracranial lesions with high specificity and sensitivity [100].

## 8. Conclusions

Mild traumatic brain injury and concussion symptoms are increasingly common problems that affect patients around the world regardless of age or gender and influence their health and quality of life. Prediction of postconcussion syndrome is beneficial in terms of proposing therapeutic options in the early phase and preventing its development. Among the available diagnostic methods, which mainly include imaging and blood examination, biomarkers are a promising prospect. An analysis of the literature shows that the abovementioned biomarkers are well-tested for severe traumatic brain injury and its consequences, but further investigations are required in the case of mTBI and the prediction of postconcussion syndrome.

## Figures and Tables

**Table 1 brainsci-14-00513-t001:** Clinical features of postconcussion syndrome. (+) symbolizes that the item is present in criteria and (−) means that the item is absent in criteria.

SYMPTOMS	ICD-10	DSM-IV
Headache	+	+
Dizziness	+	+
Fatigue	+	+
Irritability	+	+
Insomnia/sleep problems	+	+
Concentration difficulties	+	−
Memory difficulties	+	−
Intolerance of stress, emotion, alcohol	+	−
Affect changes, anxiety, depression	−	+
Personality changes	−	+

**Table 2 brainsci-14-00513-t002:** Comparison of biomarkers.

	GFAP	NSE	S100	UCH-L1	TAU PROTEIN
SPECIFICITY	78.3%	69.4%	58%	95%	83.3%
SENSITIVITY	75.6%	75%	100%	95%	100%
AUC	0.93	0.72	0.8	0.83	0.95
CUT OFF	1.35	6.45	0.38	0.21	1.75

## Data Availability

Not applicable.

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
