# Peer review of "mTBI Biological Biomarkers as Predictors of Postconcussion Syndrome—Review"

_brainsci, 2024, doi:10.3390/brainsci14050513_

Round 1
Reviewer 1 Report
Comments and Suggestions for Authors
The paper represents a comprehensive and original review on developing mild TBI predictive biomarkers. Although reviews on traumatic brain injury (TBI) theme remain a popular area, the present paper is focused on a specific clinical and molecular targets. They include predictive biomarkers of postconcussion syndrome that is commonly (but not always) developing post mild TBI. The following remarks are to be easily addressed to enchanse the quality of the review:
1. Abbreviation for mild TBI as mTBI should be appeared in abstract to fit the same abbreviation in the title. Abbreviation for postconcussion syndrome should be PCS (in capital everywhere including the Abstract). Please use abbreviation systemically, everywhere , including lines 48, 56 etc. ...
2. Imaging and neuro-phisiological markers are two briefly mentioned in the review, now any discussion on their informativity for predicting the PCS is presented.
3.Several molecular biomarkers are characterized as predictors for PCS post mTBI. However, no data on sensitivity and specificity of biomarkers and hazard ratio(HR) /odds ratio (OR) are presented. Since this part of the review seems the most important, discussion of available data on PCS and predictivity of molecular biomarkers in terms of specificity/sensitivity/HR/OR are highly desirable.
Author Response
REVIEWER 1.
- Abbreviation for mild TBI as mTBI should be appeared in abstract to fit the same abbreviation in the title. Abbreviation for postconcussion syndrome should be PCS (in capital everywhere including the Abstract).
Thank you for this comment. The corrected abbreviations you can find on whole text, for example in lines 7, 28, 56, 145, 446.
- Imaging and neuro-phisiological markers are two briefly mentioned in the review, now any discussion on their informativity for predicting the PCS is presented.
Thank you for this comemnt. We have added some informations of neurophysiological and imaging biomarkers. Additional discussion you may find in part of ‘’Definition of biomarker’’ chapter 6.
- Several molecular biomarkers are characterized as predictors for PCS post mTBI. However, no data on sensitivity and specificity of biomarkers and hazard ratio(HR) /odds ratio (OR) are presented. Since this part of the review seems the most important, discussion of available data on PCS and predictivity of molecular biomarkers in terms of specificity/sensitivity/HR/OR are highly desirable.
Thank you for this comment. Please, find our corrections in chapter 7, page 12-15.
Reviewer 2 Report
Comments and Suggestions for Authors
In the article entitled “mTBI biomarkers as predictors of postconcussion syndrome” the authors cover the literature of mTBI in the context of blood-based biomarkers. Prior to further consideration, the following points need to be addressed:
1. This sentence in the abstract seems out of place: “The biomarkers were found as a possible predictive factors that may be applied at admission to the Emergency Department”.
2. This sentence in the abstract seems out of place: “Any biomarker is confirmed to be used as a single 18 predictor in PCS.”
3. This sentence does not make sense: “Traumatic 26 Brain Injury (TBI) is a major public health problem affecting an estimated 1.5 million people each year in the United States up to 69 million people worldwide every year of 28 which 80% is mild”.
4. The manuscript needs to be significantly revised for the proper use of the English language. The way it is written now makes it very challenging to understand what the message is.
5. In the clinical picture section, expand on the GCS, the Canadian CT Head Rule and the New Orleans rule.
6. Expand on the checklist on 129 by providing more details to each section.
7. Expand on “These effects differ as a function of gender and time of injury”.
8. Expand on the Rivermead Post-Concussion Symptoms Questionnaire.
9. Line 209, instead of therapeutic, it should be theragnostic biomarker.
10. Line 233, state what was the specificity, sensitivity, AUC of NSE. Please state this information for all the blood-based biomarkers covered in the review. Also state cut-off points and present this information in a table.
11. Expand on the biology of UCHL1 and S100B with an emphasis on function.
12. It would be important to cover that GFAP and UCHL1 are FDA approved biomarkers of TBI.
13. The conclusion lacks insights into the findings covered in the review.
14. Add a section on the molecular mechanisms taking place after mTBI and represent this in a figure.
Comments on the Quality of English Language
Please, see above.
Author Response
REVIEWER 2.
- This sentence in the abstract seems out of place: “The biomarkers were found as a possible predictive factors that may be applied at admission to the Emergency Department”.
Thank you for this comment. The correct version of this sentence can be found on Abstract, line 14.
- This sentence in the abstract seems out of place: “Any biomarker is confirmed to be used as a single 18 predictor in PCS.”
Thank you for your comment. The correct version of this sentence can be found on Abstract, line 22.
- This sentence does not make sense: “Traumatic 26 Brain Injury (TBI) is a major public health problem affecting an estimated 1.5 million people each year in the United States up to 69 million people worldwide every year of 28 which 80% is mild”.
Thank you for your comment. The correct version of this sentence can be found on Introduction, line 30.
- The manuscript needs to be significantly revised for the proper use of the English language. The way it is written now makes it very challenging to understand what the message is.
Thank your for this comment. The manuscript will be corrected for proper English.
- In the clinical picture section, expand on the GCS, the Canadian CT Head Rule and the New Orleans rule.
Thank you for your comment. Due to your advice we expanded chapter of Clinical Picture. Additional information on GCS is on line 102, the Canadian CT Head Rule is on line 117, the New Orleans rule is on line 131.
- Expand on the checklist on 129 by providing more details to each section.
Thank you for this comment, but due to the fact that, probably, the manuscript that you got differs from ours, we are not able to correct sentence just by the number of line.
- Expand on“These effects differ as a function of gender and time of injury”.
Thank you for this comment. Explanation can be found in Clinical Picture chapter, line 176.
- Expand on the Rivermead Post-Concussion Symptoms Questionnaire.
Thank you for this comment. Additional information about the Rivermead Post-Concussion Symptoms Questionnaire can be found chapter of Diagnostic Criteria, line 209-225.
- Line 209, instead of therapeutic, it should be theragnostic biomarker.
Thank your for this comment. The corrected sentence can be found in Definition of biomarker, line 304.
- Line 233, state what was the specificity, sensitivity, AUC of NSE. Please state this information for all the blood-based biomarkers covered in the review. Also state cut-off points and present this information in a table.
Thank you for this comment. All the suggested information is present in chapter Need for prediction, line 422.
- Expand on the biology of UCHL1 and S100B with an emphasis on function.
Thank you for this comment. Expansion of the biology of UCHL1 and S100 can be found in chapter Definition of biomarker, line 365-374 for s100 and 351-354 for UCHL1.
- It would be important to cover that GFAP and UCHL1 are FDA approved biomarkers of TBI.
Thank you for this comment. We added that information in Definition of biomarker, line 310.
- The conclusion lacks insights into the findings covered in the review.
Thank you for this comment. Due to your advice we modified the conclusion that can be found in lines 508- 515.
Reviewer 3 Report
Comments and Suggestions for Authors
Narrative review on biomarkers in mild TBI. The authors focus mainly on biological biomarkers. Imaging and neurophysiological biomarkers are alos mentioned but play no role in this review. This should be clarified in the title, which could read „mTBI biological biomarkers as predictors of postconcussion syndrome – review.
Paragraphs 1 to 5 are too long. Please should focus on what the reader can expect from the title and cut down the introduction to few main facts.
The biomarkers in paragraph 7 should be combined with the description of the biomarkers in paragraph 6. It does not really make sense to split the information.
Author Response
REVIEWER 3.
- Paragraphs 1 to 5 are too long. Please should focus on what the reader can expect from the title and cut down the introduction to few main facts.
Thank you for this comment. Due to your advice we changed the Introduction, that can be found in lines 28-42.
- Imaging and neurophysiological biomarkers are alos mentioned but play no role in this review. This should be clarified in the title, which could read „mTBI biological biomarkers as predictors of postconcussion syndrome –
Thank you for this comment. According to your advice we changed the title.
Reviewer 4 Report
Comments and Suggestions for Authors
The authors present a manuscript describing mild TBI (mTBI) and post-concussion syndrome and the need for biomarkers to predict the effects of post-concussion syndrome. Overall the manuscript is missing several elements that are needed:
-No mention of recent mTBI guidelines for adults and children. Consider these citations:
Noah D. Silverberg Ph.D , Grant L. Iverson Ph.D. , on behalf of the ACRM Brain Injury Special Interest Group Mild TBI Task Force and the ACRM Mild TBI Definition Expert Consensus Group, ACRM Brain Injury Special Interest Group Mild TBI Task Force members, Alison Cogan Ph.D., OTRL , Kristen Dams-O’Connor Ph.D. , Richard Delmonico Ph.D. , Min Jeong P. Graf M.D. , Mary Alexis Iaccarino M.D. , Maria Kajankova Ph.D. , Joshua Kamins M.D. , Karen L. McCulloch PT, Ph.D. , Gary McKinney DHSc , Drew Nagele Psy.D. , William J. Panenka M.D. , Amanda R. Rabinowitz Ph.D. , Nick Reed Ph.D. , Jennifer V. Wethe Ph.D. , Victoria Whitehair M.D. , ACRM Mild TBI Diagnostic Criteria Expert Consensus Group, Vicki Anderson Ph.D. , David B. Arciniegas M.D. , Mark T. Bayley M.D. , Jeffrey J. Bazarian M.D., M.P.H. , Kathleen R. Bell M.D. , Steven P. Broglio Ph.D. , David Cifu M.D. , Gavin A. Davis M.B.B.S., F.R.A.C.S , Jiri Dvorak M.D., Ph.D., Ruben J. Echemendia Ph.D. , Gerard A. Gioia Ph.D. , Christopher C. Giza M.D. , Sidney R. Hinds II M.D. , Douglas I. Katz M.D. , Brad G. Kurowski M.D., M.S. , John J. Leddy M.D. , Natalie Le Sage M.D., Ph.D. , Angela Lumba-Brown M.D. , Andrew I.R. Maas M.D. , Geoffrey T. Manley M.D., Ph.D. , Michael McCrea Ph.D. , David K. Menon M.D., Ph.D. , Jennie Ponsford Ph.D. , Margot Putukian M.D. ,Stacy J. Suskauer M.D. , Joukje van der Naalt M.D., Ph.D. , William C. Walker M.D. , Keith Owen Yeates Ph.D. , Ross Zafonte D.O. , Nathan D. Zasler M.D. , Roger Zemek M.D. , The American Congress of Rehabilitation Medicine Diagnostic Criteria for Mild Traumatic Brain Injury, Archives of Physical Medicine and Rehabilitation (2023). doi:https://doi.org/10.1016/j.apmr.2023.03.036.
Silverberg, ND, Iverson, GL, on behalf of the ACRM Mild TBI Workgroup. Definition Expert Consensus Group and the ACRM Brain Injury Special Interest Group Mild TBI Task Force. Archives of Physical Medicine and Rehabilitation 2021;102:76-86.
Lumba-Brown, A., Yeates, KO., Sarmiento, K. et al. Centers for Disease Control and Prevention Guideline on the Diagnosis and Management of Mild Traumatic Brain Injury Among Children; JAMA Pediatrics. doi:10.1001/jamapediatrics.2018.2853; Published online September 4, 2018.
Areas where additional information is needed:
-Line 28 -Do you have newer citations for the reported incidence?
-Line 41 Citation 6 appears to be a clinical document, not a research study/report.
-page 2 Line 51-please describe in further detail what is mention in this sentence :"...an effect of fecklessness that is connected with senectitude..."
-page 2 Line 60-please explain this sentence ("after detailed calculations...." ) and add a citation.
-page 2 line 72-Please describe pre-injury issues and add a citation.
-page 2 line 83-please clarify and provide citation: "Patients injured in car accidents have a 54% higher risk of developing PCS that motor b
-page 2 line 86. provide further details about what you are describing..."Children due to their high activity and unwariness are common victims". Also add citation.
-page 3 line 109-111. Citation needed.
-page 3 line 135-136. "Symptoms that patients present......Citation needed.
-page 4 line 161-163. How do you know this protocol is the most commonly used? Citation needed.
-page 4 line 175. Citation is needed for sentence "Complications of mTBI is alarming".
-page 5-Line 209-210. Citation needed.
Comments on the Quality of English Language
The manuscript needs edited for English language.
Author Response
REVIEWER 4.
- No mention of recent mTBI guidelines for adults and children. Consider these citations …
Thank you for this comment. Due to your suggestion we added the guidelines in Clinical picture chapter, line 227-253, and the citations are added as well.
- Line 28 -Do you have newer citations for the reported incidence?
Thank you for this comment, but due to the fact that, probably, the manuscript that you got differs from ours, we are not able to correct sentence just by the number of line.
- Page 2 Line 51-please describe in further detail what is mention in this sentence :"...an effect of fecklessness that is connected with senectitude..."
Thank you for this comment. We expanded the sentence, can be found in Epidemiology chapter, line 47-49.
- Page 2 Line 60-please explain this sentence ("after detailed calculations...." ) and add a citation.
Thank you for this comment. We expanded the sentence, can be found in Epidemiology chapter, line 56-58.
- Page 2 line 72-Please describe pre-injury issues and add a citation.
Thank you for this comment. We expanded the sentence, can be found in Epidemiology chapter.
- Page 2 line 83-please clarify and provide citation: "Patients injured in car accidents have a 54% higher risk of developing PCS that motor.
Thank you for this comment. In Epidemiology chapter, line 78-81, you can find the explanation with citations.
- Page 2 line 86. provide further details about what you are describing..."Children due to their high activity and unwariness are common victims". Also add citation.
Thank you for this comment. In Epidemiology chapter, line 83-85, you can find the explanation with citations.
- Page 3 line 135-136. "Symptoms that patients present......Citation needed.
Thank you for this comment. In line 173 you can find added citation.
- Page 4 line 161-163. How do you know this protocol is the most commonly used? Citation needed. Page 5-Line 209-210. Citation needed.
Thank you for this comment, but due to the fact that, probably, the manuscript that you got differs from ours, we are not able to correct sentence just by the number of line.
- Page 4 line 175. Citation is needed for sentence "Complications of mTBI is alarming".
Thank you for this comment, due to your suggestion we added citation, on line 256, in Therapeutic options.
Round 2
Reviewer 2 Report
Comments and Suggestions for Authors
The AUC values in the table are pretty low. Please look into the different studies looking at those biomarkers and add the references to the table. This is perhaps the most important part of the review, so it should be covered correctly.
Comments on the Quality of English Language
The use of the English language in this article should be revised.
Author Response
Dear Reviewer,
Thank you for your valuable comment. We have added additional information to the manuscript.
Reviewer 3 Report
Comments and Suggestions for Authors
no further comments
Author Response
Dear Reviewer,
Thank you for your valuable input to our manuscript.
Reviewer 4 Report
Comments and Suggestions for Authors
The authors appear to have addressed the comments and updated the manuscript.
Comments on the Quality of English Language
A review would be helpful.
Author Response
Dear Reviewer
Thank you for your valuable comments to our work.